# Description of Daily Living Skills and Independence: A Cohort from a Multidisciplinary Down Syndrome Clinic

**DOI:** 10.3390/brainsci11081012

**Published:** 2021-07-30

**Authors:** Kavita Krell, Kelsey Haugen, Amy Torres, Stephanie L. Santoro

**Affiliations:** 1Division of Medical Genetics, Department of Pediatrics, Massachusetts General Hospital, Boston, MA 02114, USA; haugenkr@miamioh.edu (K.H.); aetorres@partners.org (A.T.); ssantoro3@mgh.harvard.edu (S.L.S.); 2Department of Pediatrics, Harvard Medical School, Boston, MA 02115, USA

**Keywords:** Trisomy 21, Down syndrome, independence, transition to adulthood, proxy-report

## Abstract

Levels of independence vary in individuals with Down syndrome (DS). We began this study to describe the current life skills in our clinic population of children and adults with DS. We collected and reviewed demographics, living situation, and life skills from an electronic intake form used in clinic procedures. Descriptive statistics for this cohort study included mean, standard deviation, and frequencies. From 2014–2020, 350 pediatric and 196 adult patients (range 0–62 years) with a first visit to the Massachusetts General Hospital Down Syndrome Program are described. Pediatric patients were most often enrolled in school, and in an inclusion setting. Adult patients were most often participating in a day program, living with family, and wanted to continue living with family in the future. Most (87%) of adults with DS communicated verbally, though fewer could use written communication (17%). Life skills of greatest importance to adolescents and adults with DS included: learning about healthy foods (35%), preparing meals (34%), providing personal information when needed (35%), and describing symptoms to a doctor (35%). Life skills for patients with DS are varied; those associated with a medical appointment, such as sharing symptoms with the doctor, could improve for greater independence.

## 1. Introduction

There are an estimated 125,461 adults with Down syndrome (DS) living in the United States [1,2]. DS is associated with co-occurring medical conditions and variable intellectual disability (ID). Individuals with DS differ in the extent to which they can complete activities of daily living; and independence, the ability to complete tasks of self-care, varies in individuals with DS [3,4,5]. A variety of factors can contribute to function, such as: cognition, health, and social factors amongst others [5]. Survey has shown that those with more current health issues were significantly less likely to be independent and social; current health issues impact communication skills [5]. Importantly, communication skills vary in individuals with DS, including studies showing: 50% of individuals with DS speaking well by age 11 years, 10 months [6], 15–45% of adults with DS using verbal communication with no difficulty [5,7], 39% of children with DS expressing with no help required [8], and 42–58% of adults understanding verbal communication [5,7]. Surveys describing one’s ability to complete various activities of daily living show a spectrum of independence in DS [5,6,7,9].

In describing the natural history of independence in individuals with DS, two studies of validated instruments emerge. One example is the Functional Independence Measure for Children (WeeFIM) questionnaire, which was used for children with DS and showed highest scores in mobility domain, and lowest in cognition domain [8]. A second example of an instrument related to independence is the Adaptive Behaviour Assessment System-II Adult (ABAS-II Adult) completed by parents and caregivers of adults with DS [10]. They found an association between increased age and lower adaptive behavior, suggesting that adults with DS may benefit from additional support in terms of their social and conceptual abilities as they age [10]. Beyond these two relevant instruments, we did not identify existing instruments which directly measure independence and are validated in adults with DS. Although studies suggest that independence and function decrease with age in adults with DS [7,9], interventions can be useful: speech training leads to increased autonomy and communication [11], and medical home access increased the odds of transition preparation and taking responsibility for health care [12].

Though the studies of independence in adulthood specific to DS are limited, lessons can be learned from the ID research literature. Natural history studies of those with ID have shown: the proportion of individuals with autism spectrum disorder (ASD) who are able to acquire a driver’s license [13] and a link between autonomy to better health and health-related quality of life (HRQOL) [14]. Some features which predict factors related to independence include: physical fitness tests (manual dexterity, balance, comfortable and fast gait speed, muscular endurance, and cardiorespiratory fitness) and changes in activities of daily living (ADLs), predictive for a decline in ability in ID [15], and a poor social network was associated with worse health outcomes in older adults in the general population [16]. A few interventions to improve independence have been studied which include aids such as videos or digital technology: video prompting improves grocery shopping in ID [17], eating aids improve independent eating in ID [18], staff trained to teach those with ID to promote self-management [19], video self-modeling improves independence [20], and the use of tech and remote support services improve independence [21].

There is need for more study of independence in individuals with DS to describe the current level of independence, as studies identified rely on data collected nearly ten years ago or longer [5,6,7]. There is a need to better understand factors, such as communication, which contribute to independence in DS. Specifically, we began this descriptive study to understand the current skills in our clinic population of individuals with DS, with the ultimate goal of using this information to (1) gain awareness and understanding on the level of independence in our clinical cohort, (2) to identify targets for future quality improvement work to improve independence, and (3) to guide future research efforts which rely on use of communication. Knowing what skills are attained could have implications on research related to surveys, interviews, and instrument development, as well as research on broader topics such as the interplay between independence level and health status.

## 2. Materials and Methods

The Mass General Hospital Down Syndrome Program (MGH DSP), is a multidisciplinary, DS specialty clinic that provides comprehensive care to 550 unique patients annually. MGH DSP offers care for infants, children, adolescents, and adults with DS through distinct clinics: Infant and Toddler Clinic (ages birth–5 years), Child Clinic (ages 5–13 years), Adolescent and Young Adult Clinic (ages 13–21 years), and Adult Clinic (ages 21 years and older). MGH DSP also offers prenatal consultation for expectant parents. The multidisciplinary team of physicians, social workers, nutritionists, and many others care for a large volume of patients with DS. This study was part of a series of ongoing projects within the program to improve the quality of our work over time.

At the MGH DSP, caregivers are asked to complete an electronic clinic intake form in advance of their loved one’s visit. The electronic intake form is shared through email, and is completed in a password-protected, Health Insurance Portability and Accountability Act (HIPAA)-secured platform which is housed through the MGH Laboratory of Computer Science; responses are stored securely on the MGH server. This electronic intake form was developed by the MGH DSP and collects a variety of pre-visit medical and lifestyle information electronically. This information is reviewed prior to the visit. We conducted a retrospective review of these electronic intake forms to evaluate patients’ activities of daily living and independence; fields reviewed included: demographics (age, sex, race, co-occurring medical conditions), school/vocational setting, living situation, communication style, daily living skills, and independence skills.

Inclusion criteria: a patient at the MGH DSP from 2014–2020, with a completed electronic intake form. Patients without an electronic intake form were not included. While the MGH DSP sees patients across the country and internationally, the electronic clinic intake form was only available in English; therefore, parents with a primary language other than English do not complete the clinic intake form. Parents of returning patients age 5 and under who are following up every 6 months are not asked to complete an additional intake form at each visit because they are seen more often than older patients. Families who were seen at MGH DSP prenatally or who gave birth at MGH and had an inpatient consult are also not asked to complete an intake form. The MGH DSP does offer a mailed paper intake for those who request it, but this data was not included in the analysis.

Descriptive statistics included: mean, standard deviation, frequencies. Data was analyzed by the patient’s age, with descriptive statistics for each subgroup reported. Data is available in de-identified, aggregate format from the author at reasonable request. This study was approved by the MassGeneral Brigham (formerly Partners) Institutional Review Board.

## 3. Results

From 2014–2020, 350 pediatric and 196 adult patients had a first visit to the MGH DSP and completed an intake form. Of these, 521 of the completed intake forms were completed by the caregiver of the individual with DS, and 25 of the intake forms were marked as self-completed by the individual with DS. Demographic details showed age ranging from 0–62 years and 46% of the patients were female. The average age of the overall cohort was 22 years, of which the average age of those completing the pediatric intake and adult intake were 10 and 35 years, respectively. The majority of the cohort self-identified as white (88%), and not Hispanic or Latino (83.3%). In the total cohort, the most common co-occurring medical condition reported was heart disease (38.6%). In the pediatric group alone, hypotonia was reported most frequently (40.8%). For adults only, heart disease remained most common, followed by thyroid disease (44.9%). (Table 1).

Daily characteristics of the sample included: pediatric patients were most often enrolled in school, and in an inclusion setting, while adult patients were most often participating in a day program, living with family, and wanted to continue living with family in the future (Table 2). Most (87%) of adults with DS communicated verbally, though fewer used written communication (17%). Assistive devices were used by both pediatric (34%) and adults (30%) with DS. Adults with DS used a smartphone (52%), read (58%), and wrote (64%). Activities of daily living which were completed by the adult with DS on his/her own ranged: 4% could cook independently, 46% bathed independently, 48% showered independently, 58% brushed teeth independently, 71% toileted independently, 74% dressed independently, and 89% fed independently (Table 2).

Independence skills for adolescents and adults with DS were ranked by importance, and those of greatest importance included: learning about healthy foods (35%), preparing meals (34%), providing personal information when needed (35%), and describing symptoms to a doctor (35%; Table 3). Skills which were most often reported as attained included: dressing self (72%), getting 7 to 8 h of sleep (58%), using a public restroom (56%), and swallowing whole pills (53%). Some skills were most often reported as not important: learning how to refill my prescriptions on my own (69%), learning what each medicine is for (61%), finding my medication list (60%). Many skills show a range in responses across the response options in the electronic intake form filled out before their clinic visit.

Among those skills ranked as really important now, but not yet attained, additional questions were asked about level of ability. Those with higher proportion showing progress to attainment included: providing personal information (71%), learning about healthy foods (71%), learning to do household chores (90%), while those with smaller proportion showing progress to attainment included: asking my doctor questions (70%), finding medication list (73%), refilling prescriptions (88%), and swallowing whole pills (70%; Table 4).

When analyzing the percent of individuals of a certain age who had attained a given skill, we saw that individuals with DS gained skills into adulthood (Table 5). Through the lifespan range, half or more of patients could get dressed on their own. Many skills were able to be completed by half or more of the individuals age 40–49 years. Some skills were less often completed through the lifespan, such as refilling prescriptions and using public transportation.

## 4. Discussion

Through retrospective review of the electronic clinic intake forms of 546 patients with DS in the MGH DSP, we found:There was great variability in activities of daily living completed by an adult with DS on their own, and most of our adults with DS used verbal communication.Dressing self, sleeping 7 to 8 h a night, using a public restroom on their own, and swallowing whole pills were the independence skills most often attained.Skills of highest importance to our patients were learning about healthy foods, preparing their own meals, communicating personal information, and describing symptoms to a doctor.

Adults with DS exhibited a great range in the type and number of activities of daily living that they were able to complete. Fewer adults with DS cooked independently, while some bathed and showered independently, and the majority were able to brush their teeth, use the restroom, and eat independently. A similar study of adults with DS separated meal preparation abilities into two categories, preparing simple meals and cooking meals; 18% said they had “a lot of difficulty” preparing simple meals like sandwiches or cereal, compared to 52.2% that said they had “a lot of difficulty” with cooking [5]. While we did not make this distinction in our electronic intake form, both rates of independence in Matthews et al. are greater than the findings in our cohort in which 4% of adults with DS were able to cook on their own. In our cohort, 46% of adults with DS bathed on their own and 48% showered on their own. A study of children with DS which looked at the level of supervision needed for bathing found similar results, with 48% requiring no help [8]. We found that 71% of adults with DS in our cohort used the restroom independently, which aligns to a previous study in which 76.4% of adults with DS used the toilet independently [5]. Lastly, we found that 89% of our adult cohort ate independently; a previous study found that 88.8% of their adult cohort could eat independently [5]. This description of activities of daily living helps to quantify the level of support needed on a daily basis for adults with DS in our cohort, which could be useful for future studies aiming at improving aspects of independence.

The preferred form of communication varies in DS; many (87%) of our adults with DS used verbal communication. Previous study found 92.3% of individuals with DS age 14–62 used verbal communication to some extent, though their results include granularity on the level of difficulty ranging from 15% who used verbal communication with no difficulty, to 18% who used verbal communication with great difficulty [7]. Additionally, in our cohort, 58% could read and 64% could write; these rates are higher than published (8.1–52.1% and 13.5–52.1%, respectively) [6,7].

The independence skills most often attained by our patients with DS age 13 and up were dressing, sleeping 7 to 8 h a night, using a public restroom on their own, and swallowing whole pills. In our cohort, 72% of patients with DS over the age of 13 dressed themselves; this rate was less than that which De Graaf et al. found, which was that 83.9% of adults studied with DS over the age of 20 were able to dress with no additional help [6]. In our cohort, 58% of our patients with DS age 13 years and older slept 7 to 8 h a night. Although studies evaluating sleep have measured sleep duration using methods such as actigraphy watches and parent report, the other studies we identified which evaluate aspects of independence have not collected information on sleeping abilities, despite the high prevalence of sleep apnea within the population [22]. Importantly, although a previous study found that 35% of adults with DS were able to take medications independently, in our cohort, more than half (53%) could swallow pills whole on their own, which has important implications for feasibility of medication administration in future clinical trials and medication adherence [5]. In addition, we found that 56% of patients in our cohort can use the public restroom on their own; this has not been evaluated in previous studies on independence, but has important implications on integrating adults with DS into the community, such as the ability to take outings in the community alone, to navigate public situations independently, and to go into settings which might require the use of a public restroom.

When planning for the future, the focus will likely be on unattained but important skills. In our cohort, the skills that were reported as most important and not yet attained were: learning about healthy foods, preparing their own meals, communicating personal information, and describing symptoms to a doctor. Those of least importance included learning to refill prescriptions, knowing what each medication is for, and finding their medication list. While other studies have looked at the level of attainment of some of these skills, none have reported on which were most important and least important for families to achieve with their loved one with DS. Behavioral support and interventions would likely best be focused on some of the unattained but important skills rather than those of lesser importance. For example, given the interest in healthy foods and meal preparation, it might be important to give additional resources to support nutritionists, feeding therapists, and related resources, like adapted cookbooks or electronic mealtime supports [23]. Resources are available online to promote independence skills [24]. Given the interest in communicating personal information, it might be helpful to create a tool for individuals with DS to practice this skill and could even be combined with the skill of describing symptoms to a doctor. One online digital healthcare tool developed by the MGH Down Syndrome Program and MGH Laboratory of Computer Science, Down Syndrome Clinic to You (DSC2U), provides families with a loved one with DS 13 years or older with a life skills checklist to identify social stories and other resources to practice and improve the life skills families select as most important right now [25]. For example, if a caregiver marks that their teenager or adult with DS would like to work on describing how they feel to their doctor, one skill we found is very important to families for independence, DSC2U will suggest specific resources to help work on this specific independence goal. If they select that they would like to work on preparing meals on their own, DSC2U will link to a cookbook designed specifically for people with DS. DSC2U can bring tailored independence resources directly to families to improve on a variety of life skills [25].

It is also interesting to note that the skills rated as unattained and least important were most often related to medical encounters. Instead of discounting these skills as not important for families, these could be growing points for physicians to support families in realizing areas their loved ones can have independence in medical settings. While a skill like refilling a prescription may feel unattainable or unnecessary at the time due to familial support, that is one area that the person with DS could exert independence with encouragement from family and providers. Having this information on the areas of greatest and least importance seems to hold opportunities for future research and intervention.

Notably, the percent of individuals of a given age who had attained given skills was high in our subgroups in their 30s and 40s. Adults in their 30s and 40s more often attained skills (in half or more of the cohort) than adolescents age 13–17 years and young adults age 18–22 years. While our cohort of adults in their 40s was still able to carry out many skills of independence, attainment seems to fall off for many of skills in the 50–59 and 60+ age groups. Differences between individuals and age groups may account for some of this variation in attainment of skill, and future studies should follow individuals longitudinally to determine if individuals can indeed gain independence skills through adulthood. However, our analysis by age subgroup provides useful evidence that adults with DS can continue to gain skills after high school and into adulthood.

One purpose of our study was to develop a baseline for future quality improvement goals for our program and to identify targets for future quality improvement work to improve independence. Our study is limited by our data collection method and the aspects of our single cohort. The electronic clinic intake form was mostly completed by parents or caregivers, but some marked that the form was self-completed by the individual with DS. So, all information collected was caregiver-reported or self-reported and not confirmed through a validated instrument or clinical notes. We cannot verify the extent of assistance, if any, with which the intake form was self-completed, therefore we are unable to distinguish whether caregivers read it to the person with DS, explained questions, or if the individual with DS completely independently completed the intake process. Although the MGH DSP sees patients across the country and internationally, non-English speakers were not included since the electronic intake form was only available in English at the time of the study. In the case of technology limitations or difficulties, caregivers were presented with the option of a mailed paper version of the form. These non-electronic, paper versions were not included in our data analysis. Additionally, our study did not distinguish between different types of heart disease; 39% had heart disease within our cohort, but this was not specifically congenital heart disease. This study reflects just one cohort of patients at our program and may not generalize to other clinics, or all individuals with DS. Future studies could expand on this review to include additional DS specialty clinics, or to track development of independence skills in our cohort over time.

In the future, it would be useful to collect self-reported data from individuals with DS of what skills are most important and most meaningful for independence, rather than caregiver-reported data. These skills can then guide future surveys on independence, future instruments to measure health, and future modifications to care. For example, the intake form did not include many questions about social activities which were asked in other studies of independence, such as working, volunteering, hanging out with friends, or playing games. If this is viewed as important to patients and their families, it could be added to better capture aspects of life which are meaningful to independence. The study provides information on the skills which are viewed as the most meaningful for independence and lays a framework for beginning to develop a measure of independence in DS. If an instrument was developed to measure independence reliably and validly, this could have important implications on future research and interventions. In comparing our results to published studies, we found that it was not easy to combine data with other independence studies given the varied methods of data collection, varied wording on surveys used, and varied approaches to reporting results as the percentage attaining a skill or the age at which a skill was attained. Developing a standard method of collection for independence skills and activities of daily living would allow researchers to expand study to include larger cohorts such as a national sample through DS-Connect [26]. Finally, with this knowledge, studies could be done to focus on how best to modify those factors which are most important for independence and daily living. Identifying effective interventions and ways to support families in building these skills could guide DS specialty clinics, researchers studying DS, and parent resource groups and advocacy organizations dedicated to individuals with DS.

## 5. Conclusions

Independence skills for patients with DS are varied; those used during a medical appointment could be improved. Skills of greatest importance should be the focus of future research and intervention.

## Figures and Tables

**Table 1 brainsci-11-01012-t001:** Demographic details of the 546 patients with Down syndrome (DS) in the Massachusetts General Hospital Down Syndrome Program (MGH DSP) with completed electronic intake forms.

	Total Cohort (*n* (%))	Pediatric Intakes (*n* (%))	Adult Intakes (*n* (%))
Male	295 (54%)	185 (53%)	110 (56%)
Race:			
White	482 (88%)	297 (85%)	185 (94%)
Black or African American	19 (4%)	14 (4%)	5 (3%)
American Indian	2 (<1%)	2 (1%)	0
Asian	52 (10%)	48 (14%)	4 (2%)
Hawaiian	1 (<1%)	1 (0.3%)	0
Other	18 (3%)	17 (5%)	1 (1%)
Ethnicity:			
Hispanic or Latino	91 (17%)	87 (25%)	4 (2%)
Co-occurring medical conditions:			
Heart disease	211 (39%)	117 (33%)	94 (48%)
Seizures	32 (6%)	13 (4%)	19 (10%)
Autism	42 (8%)	29 (8%)	13 (7%)
Dementia	16 (3%)	0	16 (8%)
Cognitive Decline	49 (9%)	12 (3%)	37 (19%)
Expressive Language Delay	125 (23%)	71 (20%)	54 (28%)
Hypotonia	190 (35%)	143 (41%)	47 (24%)
Thyroid disease	179 (33%)	91 (26%)	88 (45%)
Depression	51 (9%)	12 (3%)	39 (20%)
Anxiety	104 (19%)	50 (14%)	54 (28%)
Obsessive Compulsive Disorder (OCD)	65 (12%)	23 (7%)	42 (21%)
Attention-Deficit/Hyperactivity Disorder (ADHD)	34 (6%)	28 (8%)	6 (3%)
Post-traumatic stress disorder	9 (2%)	4 (1%)	5 (3%)
	Mean	Mean	Mean
Age (years)	22.3	10.2	35.4

**Table 2 brainsci-11-01012-t002:** Characteristics of 350 pediatric and 196 adult patients with Down syndrome (DS) in the Massachusetts General Hospital Down Syndrome Program (MGH DSP).

PEDIATRIC (0–18 Years)		*n*	%
Currently enrolled inschool		265	76
If so, placement:	Inclusion	92	35
	Partial inclusion	76	29
	Separate	73	28
	Collab	21	8
	Cotaught	9	3
	Homeschool	7	3
If so, setting:	Private	25	9
Therapies	Speech therapy at school	236	67
	Speech therapy at home	114	33
	Physical therapy at school	172	49
	Physical therapy at home	83	24
	Occupational therapy at school	201	57
	Occupational therapy at home	89	25
	Behavioral therapy at school	59	17
	Behavioral therapy at home	24	7
	Vocational therapy at school	54	15
	Vocational therapy at home	11	3
	Job coaching at school	32	9
	Job coaching at home	8	2
Has IEP		252	72
Satisfied with IEP		199	79
Uses Assistive Device		119	34
ADULT (19 years and older)			
Which best describes the current living situation	Living on his/her own	6	3
	Living on his/her own with a Supported Living Coordinator	11	6
	Living with a family member	140	71
	Living with a roommate	3	2
	Living in a group home	26	13
	Other	10	5
Where he/she would like to live in the future (check all that apply):	Live on own	16	8
	Live on own with a Supported Living Coordinator	30	15
	Live with a family member	98	50
	Live with a roommate	27	14
	Live in a group home	52	27
	Other	14	7
He/she uses an assistive device	Yes	58	30
Vocation	Currently enrolled in school	19	10
	Currently enrolled in day program	127	65
	Currently employed	96	49
He/she communicates via (check all that apply):	Verbal	170	87
	Gesture	72	37
	Sign	31	16
	Pictures	22	11
	Written	34	17
	Device	11	6
	Other	17	9
Does he/she use a smartphone or other mobile device?	Yes	101	52
Is he/she able to do on his or her own?	Feeding	175	89
	Dressing	146	74
	Bathing	90	46
	Toileting	139	71
	Showering	94	48
	Brushing teeth	114	58
	Cooking	7	4
Is he/she able to read?	Yes	114	58
Is he/she able to write?	Yes	126	64

**Table 3 brainsci-11-01012-t003:** Independence skills of 337 patients with Down syndrome age 13+ years in the MGH DSP.

	NI ^+^	TL ^‡^	RIN ^^^	SA ^§^	Attainment Was Described as:
I want to learn about the differences between healthy and unhealthy foods.	83	48	119	87	I know which foods are healthy and I try to pick the healthy foods for my meals.
I want to learn how to provide my personal information (name, emergency contact person) when needed (for example, if I get lost and a police officer asks for my name).	66	42	119	110	I am able to provide my personal information when needed without any help.
I want to learn how to describe how I am feeling to my doctor (for example, “I feel pain”, “I’m having a hard time breathing”, or “I’m coughing”)	89	65	118	65	I am always able to tell the doctor how I feel.
I want to be able to prepare my own meals.	93	96	116	32	I am able to prepare my own meals.
I want to exercise regularly.	65	41	112	119	I exercise regularly.
I want to learn how to tell the difference between a stranger and a friend.	72	38	106	121	I know how to tell the difference between a stranger and a friend.
I want to be able to bathe/shower myself.	51	30	101	155	I am able to bathe or shower without any help.
I want to understand sexual boundaries and privacy.	100	63	96	78	I believe I have a good understanding of sexual boundaries and privacy.
I want to have a plan for what I will do after finishing high school (e.g., more school, work, career goals).	113	47	93	84	I have figured out what I am doing after high school.
I want to learn how to brush my teeth on my own.	48	26	90	173	I always brush my teeth without help.
I want to learn how to call 911 if there is an emergency.	73	54	83	127	I know how to call 911 in an emergency.
I want to be able to do my laundry.	116	96	72	53	I am able to do my own laundry.
I want to learn how to do household chores.	64	46	70	157	I always help with household chores.
I want to sleep 7 to 8 h per night.	65	13	65	194	I always get 7 to 8 h of sleep at night.
I would like to name at least two adults I can ask for help when I need it.	71	30	62	174	I am able to name two adults I can go to for help.
I want to be able to use a public restroom on my own.	63	30	55	189	I am comfortable using a public restroom alone.
I want to learn how to ask my doctor questions.	130	96	54	57	I am able to ask my doctor questions on my own.
I want to be able to dress myself.	34	16	43	244	I get dressed myself.
I want to learn how to use public transportation on my own.	187	95	38	17	I am able to use public transportation when I am on my own.
I want to be able to find my medication list.	201	71	36	29	I always know where to find my medication list.
I want to learn what each of my medicines is for (for example, “I take Synthroid for my thyroid”).	205	55	33	44	I know why I take all of my medications.
I want to learn about the risks of alcohol, drugs and tobacco use.	193	46	27	71	I know how alcohol, drugs and tobacco could affect my body.
I want to learn where to find my doctor’s phone number.	192	87	26	32	I always know where to find my doctor’s phone number.
I want to learn how to swallow whole pills.	97	38	24	178	I can swallow whole pills.
I want to take my medications every day on my own.	169	59	23	86	I can take my medications with little or no help.
I want to learn how to manage my period. (for females)	44	18	22	74	I can usually manage my period without any help.
I want to learn how to find my insurance card.	178	98	19	42	I always know where my insurance card is.
I want to learn how to refill my prescriptions on my own.	233	73	17	14	I can refill my prescriptions with little or no help.

^+^ Note important. ^‡^ Not important now to me now, but I want to try later. ^^^ This is really important to me now. ^§^ Skill attained. Note: Instructions: this section should be completed by or with the patient with Down syndrome. If a particular question does not apply to the patient, choose ‘Not important’. Gray highlighting = most frequent response to each item.

**Table 4 brainsci-11-01012-t004:** Skill level if “This is really important to me now.” This corresponds to Table 3 about individuals with Down syndrome 13+ years in MGH DSP.

Topic	Response	N
Find my doctor’s phone number.	I need a lot of help finding my doctor’s phone number.	14
	I need a little help finding my doctor’s phone number.	9
Ask my doctor questions.	My parent or guardian usually asks questions for me.	38
	My parent or guardian usually reminds me to ask some questions.	16
Find my medication list.	I rely on someone else to keep the list and know my medications.	24
	I know where to find my medication list, but sometimes I forget and need some help.	9
Refill my prescriptions on my own.	I rely on my parent or guardian to refill all my medications.	15
	I sometimes help my parent or guardian call in and pick up refills of my medications.	2
Swallow whole pills.	I am unable to swallow pills and usually need to have them crushed.	17
	I am learning how to swallow a pill.	6
Tell the difference between a stranger and a friend.	I am not able to tell the difference between a stranger and a friend.	58
	I can tell the difference between a stranger and friend with help.	48
Sleep 7 to 8 h per night.	I do not usually get enough sleep at night.	41
	I am learning the importance of getting enough sleep at night.	20
Use public transportation on my own.	I do not use public transportation when I am alone.	26
	I am learning how to use public transportation without a parent or guardian.	12
Learn how to manage my period. (for females)	I need help using pads or tampons when I have my period.	11
	I am learning how to manage my period without help.	10
Brush my teeth on my own.	I need help brushing my teeth.	50
	I am learning to brush my teeth without help.	39
Use a public restroom on my own.	I need help using a public restroom.	31
	I am learning how to use a public restroom without help.	23
Find my insurance card.	I do not know where my insurance card is.	11
	I am learning where my insurance card is kept.	7
Describe how I am feeling to my doctor.	My parent or guardian usually explains how I’m feeling to the doctor.	51
	I try to describe how I am feeling and if I have trouble, my parents help me.	65
Learn what each of my medicines is for.	I do not know what any of my medications are for.	11
	I know why I am taking some of my medications.	22
Take my medications every day on my own.	I am not really sure when or how to take my medications.	8
	I am learning when to take my medications and which medications to take in the morning, noon or night (for example, I have a pill organizer that I am learning to use).	15
Provide my personal information (name, emergency contact person) when needed.	I am not able to provide any information (either verbally or nonverbally).	34
	If asked, I am able to provide some of my information or pull out an identification card that has all my information.	83
Learn about the differences between healthy and unhealthy foods.	I have trouble telling the difference between healthy and unhealthy foods.	34
	I am learning about what foods are healthy and which are not as healthy for me.	83
Learn about the risks of alcohol, drugs and tobacco use.	I do not know how alcohol, drugs and tobacco can affect me.	13
	I am learning about how alcohol, drugs and tobacco can affect me.	14
Call 911 if there is an emergency.	I do not know how to call 911 in an emergency.	35
	I am learning how to get help and call 911 in an emergency.	45
Exercise regularly.	I do not exercise regularly.	44
	I am trying to exercise regularly.	66
Learn how to do household chores.	I do not help with household chores.	7
	I am learning to help with some chores.	63
Understand sexual boundaries and privacy.	I struggle with sexual boundaries and privacy.	27
	I am learning about sexual boundaries and privacy.	65
Dress myself.	I need a lot of help getting dressed.	13
	I need a little help getting dressed.	30
Bathe/shower myself.	I need a lot of help taking a bath or with showering.	42
	I need a little help taking a bath or with showering.	58
Prepare my own meals.	Someone else prepares my meals for me.	48
	I am learning to prepare my own meals.	68
Do my laundry.	Someone else does my laundry.	22
	I am able to help with the laundry (for example, sorting or folding laundry).	49
Plan for what I will do after finishing high school.	I do not know what I am doing after high school.	29
	I am working on figuring out what to do after high school.	63
Name at least two adults I can ask for help.	I cannot name two adults I can go to for help.	19
	I am working on identifying two adults I can go to for help.	41

Note: grey highlight = more responses of minimal attainment; yellow highlight = more responses of moderate attainment.

**Table 5 brainsci-11-01012-t005:** Independence skills attained by age group of 337 patients with Down syndrome age 13+ years in the MGH DSP.

	Age Group (Years)							
Attainment Was Described as:	13–17 *n* = 83	18–22 *n* = 91	23–29 *n* = 59	30–39 *n* = 45	40–49 *n* = 24	50–59 *n* = 31	60+ *n* = 4	Total
	Skill Attained in … *n* (%)							
I get dressed myself.	60 (72)	55 (60)	49 (83)	40 (89)	19 (79)	17 (55)	2 (50)	244
I always get 7 to 8 h of sleep at night.	49 (59)	49 (54)	35 (59)	30 (67)	14 (58)	14 (45)	2 (50)	194
I am comfortable using a public restroom alone.	42 (51)	40 (44)	39 (66)	35 (78)	17 (71)	13 (42)	1 (25)	189
I can swallow whole pills.	42 (51)	39 (43)	37 (63)	25 (56)	14 (58)	16 (52)	2 (50)	178
I always brush my teeth without help.	42 (51)	36 (40)	31 (53)	30 (67)	17 (71)	15 (48)	1 (25)	173
I am able to bathe or shower without any help.	35 (42)	36 (40)	28 (47)	30 (67)	14 (58)	9 (29)	1 (25)	155
I am able to name two adults I can go to for help.	34 (41)	41 (45)	33 (56)	35 (78)	18 (75)	13 (42)	0 (0)	174
I always help with household chores.	31 (37)	39 (43)	28 (47)	28 (62)	12 (50)	16 (52)	2 (50)	157
I exercise regularly.	28 (34)	34 (37)	22 (37)	21 (47)	7 (29)	7 (23)	0 (0)	119
I know how to call 911 in an emergency.	25 (30)	29 (32)	20 (34)	29 (64)	14 (58)	9 (29)	0 (0)	127
I know how to tell the difference between a stranger and a friend.	21 (25)	28 (31)	21 (36)	22 (49)	15 (63)	13 (42)	1 (25)	121
I know which foods are healthy and I try to pick the healthy foods for my meals.	19 (23)	16 (18)	13 (22)	22 (49)	11 (46)	6 (19)	0 (0)	87
I am able to provide my personal information when needed without any help.	18 (22)	23 (25)	26 (44)	21 (47)	13 (54)	9 (29)	0 (0)	110
I can take my medications with little or no help.	15 (18)	23 (25)	16 (27)	14 (31)	11 (46)	5 (16)	0 (0)	86
I know how alcohol, drugs and tobacco could affect my body.	14 (17)	13 (14)	13 (22)	19 (42)	7 (29)	5 (16)	0 (0)	71
I am always able to tell the doctor how I feel.	13 (16)	13 (14)	12 (20)	11 (24)	7 (29)	8 (26)	0 (0)	65
I am able to ask my doctor questions on my own.	11 (13)	16 (18)	8 (14)	12 (27)	5 (21)	5 (16)	0 (0)	57
I know why I take all of my medications.	11 (13)	10 (11)	10 (17)	8 (18)	3 (13)	2 (6)	0 (0)	44
I always know where to find my doctor’s phone number.	7 (8)	6 (7)	6 (10)	8 (18)	2 (8)	3 (10)	0 (0)	32
I believe I have a good understanding of sexual boundaries and privacy.	6 (7)	15 (16)	16 (27)	26 (58)	8 (33)	7 (23)	0 (0)	78
I have figured out what I am doing after high school.	3 (4)	16 (18)	25 (42)	26 (58)	9 (38)	4 (13)	0 (0)	84
I always know where to find my medication list.	3 (4)	5 (5)	4 (7)	9 (20)	6 (25)	2 (6)	0 (0)	29
I am able to do my own laundry.	3 (4)	9 (10)	9 (15)	16 (36)	9 (38)	6 (19)	0 (0)	53
I am able to prepare my own meals.	2 (2)	6 (7)	3 (5)	8 (18)	7 (29)	6 (19)	0 (0)	32
I always know where my insurance card is.	1 (1)	7 (8)	8 (14)	13 (29)	3 (13)	8 (26)	1 (25)	42
I can refill my prescriptions with little or no help.	1 (1)	3 (3)	1 (2)	4 (9)	4 (17)	1 (3)	0 (0)	14
I am able to use public transportation when I am on my own.	1 (1)	1 (1)	3 (5)	8 (18)	4 (17)	0 (0)	0 (0)	17
I can usually manage my period without any help.	24	14	18	10	6	1	0	74

Note: this section should be completed by or with the patient with Down syndrome. If a particular question does not apply to the patient, choose ‘Not important’. Green ≥ 50% attained; Yellow = 25–49% attained; Red = 10–24% attained; Gray < 10% attained.

## Data Availability

The data presented in this study are available on request from the corresponding author. The data are not publicly available because we did not notify survey respondents of data sharing at time of completing the survey, therefore, they did not explicitly consent to data sharing.

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
