# Peer review of "Description of Daily Living Skills and Independence: A Cohort from a Multidisciplinary Down Syndrome Clinic"

_brainsci, 2021, doi:10.3390/brainsci11081012_

Round 1
Reviewer 1 Report
Problem Statement: Life skills studies in the DS populations is dated. There is need for more study of independence in individuals with DS to describe the current level of independence. There is a need to develop a baseline for future quality improvement goals for their program.
The authors describe the current life skills in their clinic population of children and adults with DS in a larger DS practice. Using descriptive statistics, they determine that life skills like learning about healthy foods, preparing meals, providing personal information when needed, and describing symptoms to a doctor are important. Life skills for patients with DS are varied; New targets were identified- Life skills associated with a medical appointment, such as sharing symptoms with the doctor may need more training and improvement in general to better support independent life for those with DS.
Introduction:
(line 40) the WeeFIM system is mentioned for children with DS. Efforts were made to describe the WeeFIM system and correlation with age with individuals with DS. A small chart of this clinics results using this cohort data might be useful to validate the claim for use. Consider Age of life (y-axis continuous) vs categorical group columns.
Results:
Table 1 demographics and Table 2 Characteristic answers and Table 4 “Skill level important to me” are very good. Table 3 is overall fine but could be restructured to make it easier to read. Aim to eliminate the large space voids in the cells and more space for the larger phrases if possible. Table 5 with age cross section breakdowns is also very good.
Discussion:
(line 240) It was very thoughtful to include one online digital heathcare tool DSC2U. For an outside clinic it migh be useful to describe more on how that tool has or could improve a specific life skill.
(line 255, paragraph) The author should further discuss Table 5. What would the authors say about skills in ages 40-49, 50-59 & 60+.
This reviewer appreciates the care given to analyze potential shortfalls of the study and/or translatability to other clinics etc. Highlighting “self-reporting” vs “care-giver assisted reporting” is also a very interesting prospect that could be an enabler of independence for many with DS.
Reviewer 2 Report
Congratulations. Excellent and most useful research. Although as you point out there is still work to do to improve the living skills of the persons with Down syndrome. Present-day indications are globally positive. I would have expected less.
- The question addressed as clearly indicated in the paper (title etc.) is the quality of daily living and adaptation in persons with Down syndrome living the open community.
- Very few studies have addressed this question with such precision and systematicity. The resulting indications are globally positive but there is still some work to do in order to improve the situation and the authors have identified particular areas whereby additional progresses are needed.
- The paper is well written and reads well.
- The conclusions are consistent (see point 2 here).
